# Emergency SARS-CoV-2 Variants of Concern: Novel Multiplex Real-Time RT-PCR Assay for Rapid Detection and Surveillance

Hsing-Yi Chung,[a] Ming-Jr Jian,[a] Chih-Kai Chang,[a] Jung-Chung Lin,[b] Kuo-Ming Yeh,[b] Chien-Wen Chen,[c] Shan-Shan Hsieh,[a] Kuo-Sheng Hung,[d] Sheng-Hui Tang,[a] Cherng-Lih Perng,[a] Feng-Yee Chang,[b] Chih-Hung Wang,[e] [ID] Hung-Sheng Shang[a]

[a]Division of Clinical Pathology, Department of Pathology, Tri-Service General Hospital, National Defense Medical Center, Taipei, Taiwan, Republic of China

[b]Division of Infectious Diseases and Tropical Medicine, Department of Medicine, Tri-Service General Hospital, National Defense Medical Center, Taipei, Taiwan, Republic of China

[c]Division of Pulmonary and Critical Care Medicine, Department of Medicine, Tri-Service General Hospital, National Defense Medical Center, Taipei, Taiwan, Republic of China

[d]Center for Precision Medicine and Genomics, Tri-Service General Hospital, National Defense Medical Center, Taipei, Taiwan, Republic of China

[e]Department of Otolaryngology-Head and Neck Surgery, Tri-Service General Hospital, National Defense Medical Center, Taipei, Taiwan, Republic of China

Hsing-Yi Chung and Ming-Jr Jian contributed equally to this article. Author order was determined alphabetically and in order of increasing seniority.

**ABSTRACT** Coronavirus disease 2019 (COVID-19), caused by severe acute respiratory syndrome coronavirus 2 (SARS-CoV-2), has spread worldwide. Many variants of SARS-CoV-2 have been reported, some of which have increased transmissibility and/or reduced susceptibility to vaccines. There is an urgent need for variant phenotyping for epidemiological surveillance of circulating lineages. Whole-genome sequencing is the gold standard for identifying SARS-CoV-2 variants, which constitutes a major bottleneck in developing countries. Methodological simplification could increase epidemiological surveillance feasibility and efficiency. We designed a novel multiplex real-time reverse transcriptase PCR (RT-PCR) to detect SARS-CoV-2 variants with S gene mutations. This multiplex PCR typing method was established to detect 9 mutations with specific primers and probes (ΔHV 69/70, K417T, K417N, L452R, E484K, E484Q, N501Y, P681H, and P681R) against the receptor-binding domain of the spike protein of SARS-CoV-2 variants. *In silico* analyses showed high specificity of the assays. Variants of concern (VOC) typing results were found to be highly specific for our intended targets, with no cross-reactivity observed with other upper respiratory viruses. The PCR-based typing methods were further validated using whole-genome sequencing and a commercial kit that was applied to clinical samples of 250 COVID-19 patients from Taiwan. The screening of these samples allowed the identification of epidemic trends by time intervals, including B.1.617.2 in the third Taiwan wave outbreak. This PCR typing strategy allowed the detection of five major variants of concern and also provided an open-source PCR assay which could rapidly be deployed in laboratories around the world to enhance surveillance for the local emergence and spread of B.1.1.7, B.1.351, P.1, and B.1.617.2 variants and of four Omicron mutations on the spike protein (ΔHV 69/70, K417N, N501Y, P681H).

**IMPORTANCE** COVID-19 has spread globally. SARS-CoV-2 variants of concern (VOCs) are leading the next waves of the COVID-19 pandemic. Previous studies have pointed out that these VOCs may have increased infectivity, have reduced vaccine susceptibility, change treatment regimens, and increase the difficulty of epidemic prevention policy. Understanding SARS-CoV-2 variants remains an issue of concern for all local government authorities and is critical for establishing and implementing effective public health measures. A novel SARS-CoV-2 variant identification method based on a multiplex real-time RT-PCR was developed in this study. Five SARS-CoV-2 variants (Alpha, Beta, Gamma, Delta, and Omicron) were identified simultaneously using this method. PCR typing can provide rapid testing results with lower cost and higher feasibility, which is well within

**Ad Hoc Peer Reviewer** [ID] Yao Jiang, Lanzhou University Second Hospital; [ID] Luciana Costa, Universidade Federal do Rio de Janeiro

Address correspondence to Hung-Sheng Shang, iamkeith001@gmail.com, or Chih-Hung Wang, chw@ms3.hinet.net.

The authors declare no conflict of interest.

the capacity for any diagnostic laboratory. Characterizing these variants and their mutations is important for tracking SAR-CoV-2 evolution and is conducive to public infection control and policy formulation strategies.

**KEYWORDS** ΔHV 69/70, E484K, E484R, K417N, K417T, L452R, N501Y, P681H, P681R, SARS-CoV-2, VOC genotyping, epidemiological surveillance, SARS-CoV-2

As of January 2022, more than 340 million cases of COVID-19 have been reported and are associated with over 5 million deaths globally (1) (https://covid19.who.int/). Mutations arise as a natural occurrence of viral replication. When a newly arising mutation confers a competitive advantage with respect to viral replication, transmission, or escape from immunity, that mutation increases in prevalence in the overall virus population (2, 3). Importantly, new SARS-CoV-2 variants of concern (VOCs) may enhance virus transmissibility and/or disease severity, as well as diagnostic and/or treatment failure (4–6). SARS-CoV-2 lineages carrying the amino acid substitution N501Y spread rapidly in the United Kingdom in late autumn of 2020 (7). SARS-CoV-2 lineage with multiple spike mutations in South Africa was first reported in October 2020 and then spread across Western and Eastern Cape provinces, as the B.1.351 variant (8, 9). P.1, referred to as the Gamma variant, was first detected in early March 2020 and spread in Manaus, Brazil (10). B.1.617.2 sublineage, or Delta, was first detected in India in December 2020 and rapidly became the dominant variant in India (9). Currently, SARS-CoV-2 variants are detected mainly by whole-genome sequencing (WGS) (2, 11). Although genome sequencing is the gold standard for identifying SARS-CoV-2 variants, resource and capacity constraints can limit the number of samples that can be sequenced. Therefore, establishing a rapid, accurate, economic, and multisite detection method for SARS-CoV-2 variant identification is an urgent technical need for SARS-CoV-2 infection prevention and control worldwide.

Here, we evaluated the analytical and clinical performance of our real-time reverse transcription PCR (RT-PCR) typing method to qualitatively detect SARS-CoV-2 RNA from wild type (non-VOC) or VOCs (B.1.1.7, B.1.351, P.1, and B.1.617.2), particularly with respect to the VOC B.1.617.2 lineage that is driving the third wave of the Taiwan COVID-19 epidemic. This study also provided preliminary results indicating that the VOC PCR typing method was able to detect the fifth VOC, Omicron (B.1.1.529), which has rapidly spread across the globe.

## RESULTS

**VOC/VOI genotyping assay design.** VOCs or variants of interest (VOIs) were collected from five dominant SARS-CoV-2 strains (Delta-lineage B.1.617.2+AY.x, Alpha-lineage B.1.1.7+Q.x, Beta-lineage B.1.351+B.1.351.2+B.1.351.3, GammaBeta-lineage P.1+P.1.x, and Omicron-lineage B.1.1.529) from GIASIAD, which contains the sequences of more than 4,700,000 isolates identified since 2019 (Fig. 1). Based on these VOCs and VOIs, comparing single nucleotide polymorphisms (SNPs) in the spike region to those in Wuhan-hu-1, nine different TaqMan SNP genotyping assays were designed to detect mutations present in clinical specimens. Locations of forward and reverse sequence-specific primers and dye-labeled probes for the TaqMan SNP genotyping assays are shown in Fig. 2.

**Analytical sensitivity of lab-developed VOC real-time RT-PCR assay.** The limit of detection (LoD) of our lab-developed VOC RT-PCR assay was determined by testing 20 replicates of SARS-CoV-2 gene fragment controls that were serially diluted 2-fold. Using the LightCycler 96 Thermocycler (Roche, Mannheim, Germany), the LoD obtained from the 20 replicate tests was 30 copies/$\mu$L for the mutant N501Y and 60 copies/$\mu$L for the mutants ΔHV 69/70, K417N, E484K, L452R, and P681R (Table 1). In the AIO 48 open system (LabTurbo, New Taipei City, Taiwan), the LoD had the same performance (Table 1). A result was considered positive if the amplification curve crossed the threshold line within 31 cycles (threshold cycle [$C_T$] value < 31) (Table S1).

(A)

| Variant | ΔHV69-70 | K417N | K417T | L452R | E484K | E484Q | N501Y | D614G | P681H | P681R |
|---------|----------|-------|-------|-------|-------|-------|-------|-------|-------|-------|
| Omicron | 233,426 | 78,817 | 8 | 1,615 | 4 | 9 | 214,193 | 240,224 | 239,308 | 166 |
| Delta | 5,896 | 7,381 | 85 | 3,886,002 | 1,542 | 10,327 | 1,664 | 3,986,848 | 1,229 | 3,978,721 |
| Alpha | 1,098,270 | 155 | 22 | 463 | 3,539 | 397 | 1,131,975 | 1,152,771 | 1,149,328 | 2,322 |
| Beta | 22 | 36,005 | 2 | 57 | 34,462 | 0 | 34,727 | 39,301 | 53 | 48 |
| Gamma | 392 | 54 | 114,462 | 107 | 114,088 | 1 | 114,418 | 119,601 | 6,049 | 443 |

(B)

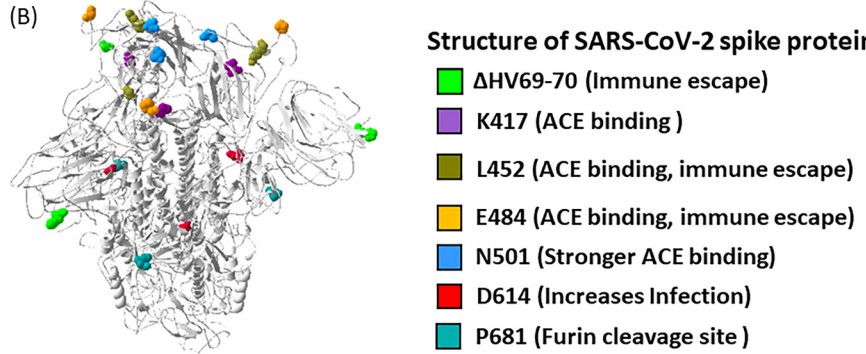

**Structure of SARS-CoV-2 spike protein**

- 🟩 **ΔHV69-70 (Immune escape)**
- 🟪 **K417 (ACE binding )**
- 🟫 **L452 (ACE binding, immune escape)**
- 🟧 **E484 (ACE binding, immune escape)**
- 🟦 **N501 (Stronger ACE binding)**
- 🟥 **D614 (Increases Infection)**
- 🟦 **P681 (Furin cleavage site )**

**FIG 1** Five variants of interest (VOIs) or variants of concern (VOCs) of SARS-CoV-2 strains collected from GISAID. (A) VOI (K417T and P681H) and VOC (ΔHV69-79, K417N, L452R, E484K, E484Q, N501Y, and P681R). (B) VOI and VOC positions in SARS-CoV-2 spike protein structure and related biological effects. Lime, ΔHV69-70; violet, K417; olive, L452; orange, E484; dodger blue, N501; red, D614; light sea green, P681.

**Analytical specificity of lab-developed VOC real-time RT-PCR assay.** Targets to differentiate between B.1.1.7, B.1.351, P.1, B.1.1.672, and Omicron (B.1.1.529) were analyzed over 4,700,000 SARS-CoV-2 genomes on GISAID. *In silico* analyses showed high specificity of the assays. The analytical specificity of each assay was determined by testing a panel of pathogens which included influenza A, influenza B, rhinovirus, enterovirus, parainfluenza virus subtypes 1 to 3, adenovirus, coronavirus HKU1, and COVID-19-negative samples. All test results were found to be highly specific for our intended targets, with no cross-reactivity observed with other upper respiratory viruses or COVID-19-negative samples ($n = 500$).

**Clinical performance of lab-developed VOC real-time RT-PCR assay.** We analyzed 250 clinically positive samples in this study, which were confirmed by the Taiwan CDC central laboratory. Figure 3 depicts the VOC assay interpretation in this study design. These positive samples were further identified as wild type (non-VOC) or VOCs (B.1.1.7, B.1.351, P.1, and B.1.617.2). To confirm that the results were consistent with those of rapid detection of the SARS-CoV-2 VOCs, presumptive cases were also assessed by whole-genome sequencing, and the lineages were confirmed using GISAID software. To verify the accuracy of our lab-developed VOC typing assay, we also used commercial VirSNiP SARS-CoV-2 mutation assays for strain surveillance and compared the VOC typing results (Table 2). The findings showed that our novel RT-PCR method exhibited better analytic sensitivity and lower $C_T$ values (31 to 34) than those of the commercial kit.

## DISCUSSION

Nucleic acid-amplification test assays for SARS-CoV-2 VOCs typing employ real-time RT-PCR, which takes anywhere from less than one to a few hours, thus shortening the turnaround time compared with that required for whole-genome sequencing. In this study, we determined the analytical sensitivity of the VOC genotyping assay. Similar LoD values were obtained with two different PCR systems (AIO 48 and LC96), so the assay could be implemented in other diagnostic laboratories using their own resources, equipment, and personnel. We also examined the clinical performance using

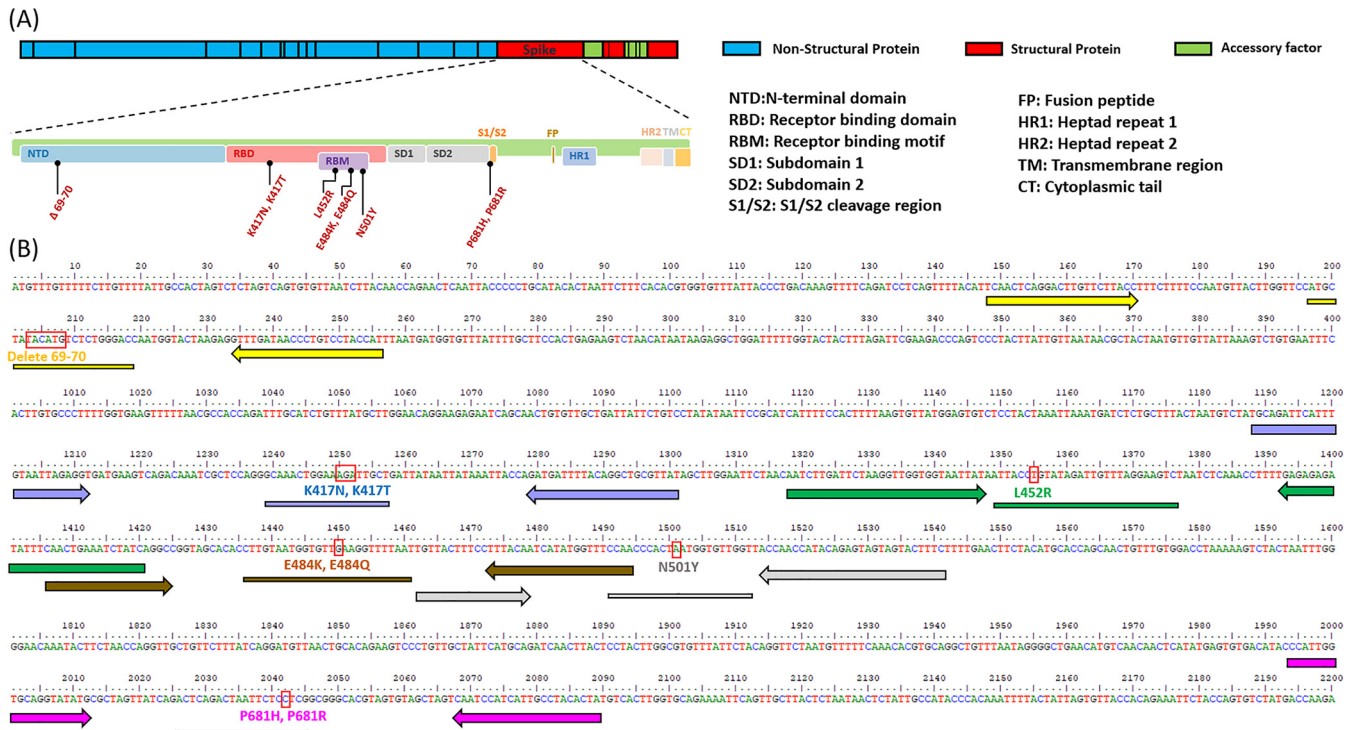

**FIG 2** VOI/VOC sequence-specific forward and reverse primers and probes designed for TaqMan SNP genotyping assays. (A) Spike gene position in SARS-CoV-2 genome and its amino acid functional annotation. (B) Specific primer-pair (arrow bar) and probe (long bar) positions in spike gene. Red boxes show detected mutations. Yellow, ΔHV69-70; light violet, K417N and 417T; green, L452R; brown, E484K and E484Q; gray, N5011Y; and pink, P681H and P681R.

COVID-19 patient samples from northern Taiwan. Our findings indicate that the novel RT-PCR method of SARS-CoV-2 VOCs typing is able to identify and differentiate the major four VOCs of SARS-CoV-2 (B.1.1.7, B.1.351, P.1, and B.1.617.2). Our assay is faster and simpler than whole-genome sequencing, which is the gold standard method. With the emergence of the B.1.1.7 lineage, Taiwan confronted a COVID-19 flare-up in May 2021 (12) and later in June 2021. The first imported B.1.617.2 cases were reported by Taiwan Central Epidemic Command Center (CECC), which later led to outbreaks of community transmission (13). To meet this challenge, assays for the detection of SARS-CoV-2 VOCs were designed and implemented in our hospital. These RT-PCR assays

**TABLE 1** Assessment of limit of detection for VOC RT-PCR[a]

| Thermocycler | Gene target/ fluorescent dye | No. of replicates detected as positive/no. of replicates tested at indicated RNA control copy no. (percentage) | | | | | |
|---|---|---|---|---|---|---|---|
| | | 240 | 120 | 60 | 30 | 15 | 7.5 |
| LightCycler 96 | N501Y/FAM | 20/20 (100) | 20/20 (100) | 20/20 (100) | 19/20 (95) | 17/20 (85) | 10/20 (50) |
| | ΔHV 69/70/FAM | 20/20 (100) | 20/20 (100) | 20/20 (100) | 17/20 (85) | 12/20 (60) | ND |
| | E484K/HEX | 20/20 (100) | 20/20 (100) | 20/20 (100) | 17/20 (85) | 12/20 (60) | ND |
| | K417N/FAM | 20/20 (100) | 20/20 (100) | 20/20 (100) | 17/20 (85) | 12/20 (60) | ND |
| | L452R/FAM | 20/20 (100) | 20/20 (100) | 20/20 (100) | 17/20 (85) | 12/20 (60) | ND |
| | P681R/HEX | 20/20 (100) | 20/20 (100) | 20/20 (100) | 17/20 (85) | 12/20 (60) | ND |
| LabTurbo AIO 48 | N501Y /FAM | 20/20 (100) | 20/20 (100) | 20/20 (100) | 19/20 (95) | 16/20 (80) | 11/20 (55) |
| | ΔHV 69/70/FAM | 20/20 (100) | 20/20 (100) | 20/20 (100) | 17/20 (85) | 12/20 (60) | ND |
| | E484K/HEX | 20/20 (100) | 20/20 (100) | 20/20 (100) | 17/20 (85) | 12/20 (60) | ND |
| | K417N/FAM | 20/20 (100) | 20/20 (100) | 20/20 (100) | 17/20 (85) | 12/20 (60) | ND |
| | L452R/FAM | 20/20 (100) | 20/20 (100) | 20/20 (100) | 17/20 (85) | 12/20 (60) | ND |
| | P681R/HEX | 20/20 (100) | 20/20 (100) | 20/20 (100) | 17/20 (85) | 12/20 (60) | ND |

[a]ND: Not detected.

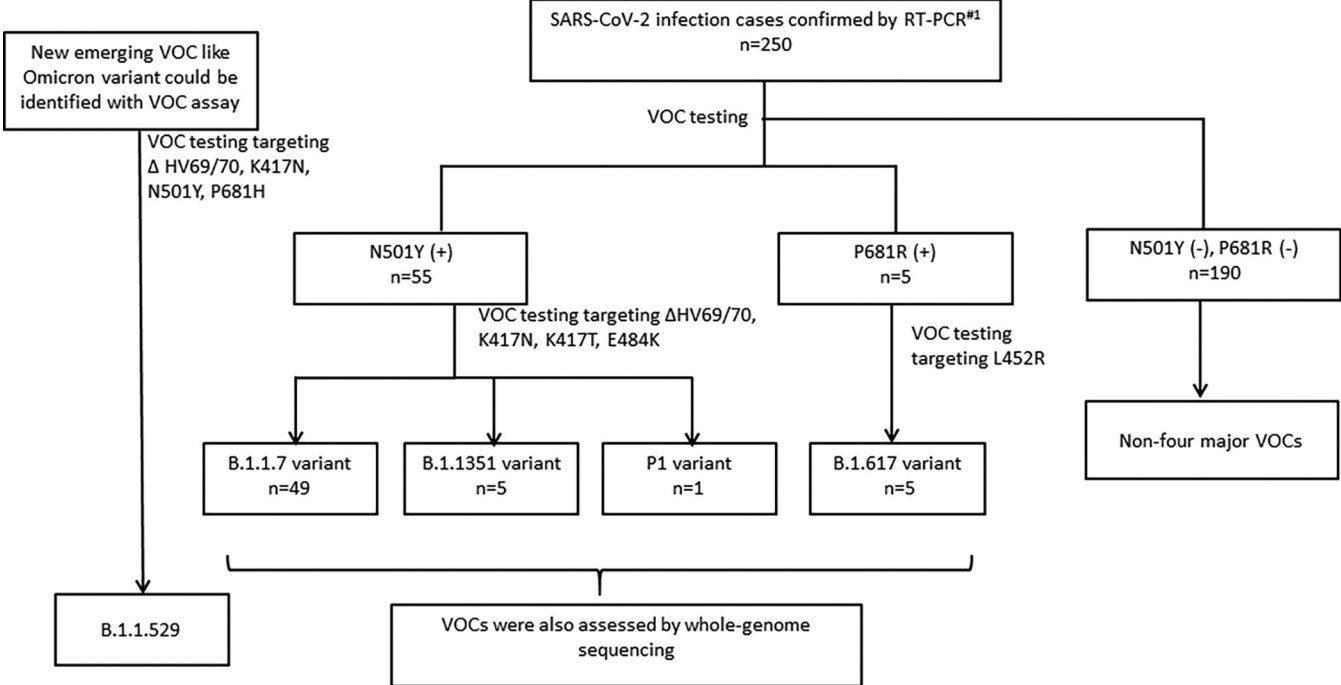

**FIG 3** Flow chart of interpretation of the VOC assays.

targeting the nine mutation sites allowed the rapid identification of B.1.1.7, B.1.617.2, and other VOCs, which helped the government rapidly follow up and reformulate more efficient strategies to deal with the emergence of VOCs.

Relying on genome sequencing platforms for identifying SARS-CoV-2 VOCs requires expensive instruments and ample resources, which constitutes a major bottleneck in developing countries. The VOC screening assay design implemented in our study shortens the time required to obtain results, and hence the number of laboratories capable of testing for SARS-CoV-2 VOCs should be expanded.

There were some limitations to our study. First, our designed novel RT-PCR method focuses on identifying currently well-known VOCs. Whole-genome sequencing is thus indispensable to detect new emerging variants when the PCR method fails to obtain results. Second, this PCR assay can be used to screen only known SARS-CoV-2-positive clinical samples for the presence of the major VOCs.

Overall, the rapid implementation of mutation-specific nucleic acid tests to detect four major VOCs should aid public health measures in identifying these emerging

**TABLE 2** Result of SARS-CoV-2 VOC testing by two methodologies[a]

| | | Number of detectable samples of variant by methodology: | | | | | | | |
|---|---|---|---|---|---|---|---|---|---|
| | | B.1.1.7 | | B.1.351 | | P.1 | | B.1.617 | |
| Spike | Spike variation | qPCR | TIB | qPCR | TIB | qPCR | TIB | qPCR | TIB |
| N501 | N501Y | 49 | 43[b] | 5 | 5 | 1 | 1 | | |
| HV69/70 | ΔHV 69/70 | 49 | 43[b] | | | | | | |
| E484 | E484K | | | 5 | 5 | 1 | 1 | | |
| L452 | L452R | | | | | | | 5 | 3[b] |
| K417 | K417N | | | 5 | 5 | | | | |
| P681 | P681H | 49 | 43[b] | | | | | | |
| P681 | P681R | | | | | | | 5 | 3[b] |

[a]TIB, VirSNiP SARS-CoV-2 mutation assays for strain surveillance; qPCR, novel RT-PCR detection of SARS-CoV-2 VOC.
[b]Reduced number of detectable samples compared to the other methodology; $C_T$ value ranged from 31 to 34.

threats. In addition, these VOC assays can provide reliable information at lower viral loads than whole-genome sequencing as described previously (14). We also observed in our own studies that when lower viral loads of SARS-CoV-2 samples were used, we often failed to generate whole-genome sequencing data.

Modern transportation plays a key role in the spread of SARS-CoV-2 and new variants (15), providing a reminder of the necessity of global epidemiological surveillance. Fully vaccinated breakthrough infections have been reported in many countries, involving different SARS-CoV-2 variants (16–20). Travel restrictions can be lifted when high vaccination coverage is reached only if vaccines remain highly effective against VOCs (21). This also highlights the importance of VOC surveillance for government and public health departments and presses for increased vaccination coverage.

While the manuscript was being written, a new SARS-CoV-2 variant, B.1.1.529, rose rapidly in Gauteng province in South Africa (22). Later, B.1.1.529 was announced as a VOC called Omicron by the World Health Organization (23). Our method accurately detected four Omicron mutations on the spike protein (ΔHV 69/70, K417N, N501Y, P681H) with one pass. Using our novel RT-PCR VOC typing technique, we successfully detected the Omicron variant B.1.1.529 (including BA.1 and BA.2) in accordance with whole-genome sequencing (Table S2). It is clear that our RT-PCR assays can not only detect single mutations for identifying currently known VOCs but also detect emerging VOCs. The preliminary conclusion is that our method can rapidly detect B.1.1.529 without sequencing the entire viral genome.

**Conclusions.** This study describes the development and validation of multiplex real-time RT-PCR assays to detect major mutations of VOCs. Our method demonstrated greater flexibility and more accessibility than those of other diagnostic laboratories and is the first of its kind to detect the five major VOCs, since SARS-CoV-2 VOCs emerged as a global threat worldwide. The molecular-diagnostics protocol developed by us simultaneously tested for B.1.1.7, B.1.351, P.1, B.1.617.2, and B.1.1.529. This should significantly benefit hospitals and health departments and enable effective management and epidemiological surveillance of the COVID-19 pandemic.

## MATERIALS AND METHODS

**Study design and clinical specimens.** This study was registered on February 8, 2021 and approved by the Tri-Service General Hospital Institutional Review Board (approval number C202005041). We tested 750 deidentified nasopharyngeal swab specimens collected from patients confirmed as having COVID-19, including 250 positive and 500 negative specimens. Residual viral transport medium was stored at −80℃. The sample collection periods were between June 2020 and December 2021.

**RT-PCR testing for SARS-CoV-2 detection.** Sample testing for SARS-CoV-2 was done as described previously (24, 25). Briefly, SARS-CoV-2 RT-PCR testing was performed using the LabTurbo AIO 48 system (LabTurbo, New Taipei City, Taiwan) for detecting the SARS-CoV-2 $N1$ and $E$ genes. A $C_T$ value of <35 was defined as a positive result for the pathogen. Each sample had an internal control (RNase $P$ gene). The external control comprised RNA spike-in mix as the positive control and diethyl pyrocarbonate (DEPC) water as the negative control.

**Novel RT-PCR assay design.** SARS-CoV-2 genomes were downloaded from the GISAID database (26). Primers and probes were designed using the consensus sequences obtained from the sequence alignment for the spike protein region of SARS-CoV-2 (Table 3).

**RT-PCR detection of SARS-CoV-2 VOC.** The positive SARS-CoV-2 specimens were screened by six multiplex RT-PCR assays. A 20-$\mu$L reaction mixture was made up of 5 $\mu$L of RNA and 15 $\mu$L PCR master mix containing the primer/probe mixture. The master mix was from Luna one-step RT-PCR kit (New England Biolabs). The assay for detection of the N501Y mutation was adapted from a multiplex RT-PCR assay for detection of spike mutants in SARS-CoV-2. The primer and probe overlap the sequences that contain mutant amino acids N501Y and its wild type N501N (Table 3). The assays were performed under the following conditions: reverse transcription at 55℃ for 8 min and initial denaturation at 95℃ for 1 min, 45 cycles at 95℃ for 10 s, and 58℃ for 20 s. When the RNA contains the sequence perfectly matching the probe, there will be a signal generated in the channel of the designed sequence. For example, positivity for mutant N501Y is when the FAM channel has a signal, and negativity for N501Y is when the HEX channel has a positive signal. In addition, three separate assays were designed to detect spike mutations and wild type, E484K versus E484E, L452R versus L452L, and K417N versus K417K. For the ΔHV 69/70 assay, positive fluorescence signal means RNA with wild-type sequence. Using a similar strategy, a primer/probe set that targets L452 and P681 was performed under the following conditions: reverse transcription at 55℃ for 8 min and initial denaturation at 95℃ for 10 min, 45 cycles at 95℃ for 15 s and 64℃ for 30 s. All six multiplex RT-PCR assays were examined on two different real-time PCR instruments, the Roche LightCycler 96 System and the LabTurbo AIO 48 open system.

**TABLE 3** Primer sequences of the multiplex RT-PCR assay

| Spike protein | Nucleotide | Amino acid | Forward primer 5'-3' | Reverse primer 5'-3' | Probe 5'-3' | Fluorescence |
|---|---|---|---|---|---|---|
| N501 | A23063A | N501N | TGTTACTTTCCTTTACAATCATATGGTTT | GAAAGTACTACTACTCTGTATGGTTGGTA | 5'-/5Cy5/CCAACCCAC/TAO/TAATGGTGTTGG/3IABkFQ/-3' | Cy5 |
| | A23063T | N501Y | TGTTACTTTCCTTTACAATCATATGGTTT | GAAAGTACTACTACTCTGTATGGTTGGTA | 5'-/56-FAM/CCAACCCAC/ZEN/TTATGGTGTTGG/3IABkFQ/-3' | FAM |
| Δ69-70 | 21765-21770 | 69-70 | TCAACTCAGGACTTGTTCTTAC | TGGTAGGGACAGGGTTATCAAAC | 5'-/56-FAM/GTCCCAGAG/ZEN/ACATGTATAGCAT/3IABkFQ/-3' | FAM |
| E484 | G23012G | E484E | CAACTGAAATCTATCAGGC | TTGGAAACCATATGATTGTAAAG | 5'-/5Cy5/GTAATGGTG/TAO/TTGAAGGTTT/3IABkFQ/-3' | Cy5 |
| | G23012C | E484Q | CAACTGAAATCTATCAGGC | TTGGAAACCATATGATTGTAAAG | 5'-/56-FAM/ATGGTGTTC/ZEN/AAGGTTTTAAT/3IABkFQ/-3' | FAM |
| | G23012A | E484K | CAACTGAAATCTATCAGGC | TTGGAAACCATATGATTGTAAAG | 5'-/5HEX/CTTGTAATG/ZEN/GTGTTAAAGGT/3IABkFQ/-3' | HEX |
| K417 | A22812A | K417K | TGCAGATTCATTTGTAATTAGAGG | ATAACGCAGCCTGTAAAATCATC | 5'-/5Cy5/GCAAACTGG/TAO/AAAGATTGCT/3IABkFQ/-3' | Cy5 |
| | A22812C | K417T | TGCAGATTCATTTGTAATTAGAGG | ATAACGCAGCCTGTAAAATCATC | 5'-/5HEX/GCAAACTGG/ZEN/AACGATTGCT/3IABkFQ/-3' | HEX |
| | G22813T | K417N | TGCAGATTCATTTGTAATTAGAGG | ATAACGCAGCCTGTAAAATCATC | 5'-/56-FAM/GCAAACTGG/ZEN/AAATATTGCT/3IABkFQ/-3' | FAM |
| L452 | T22917T | L452L | TGATAGATTTCAGTTGAAATATCTCTCTCA | AATCTTGATTCTAAGGTTGGTGGTAATTAT | 5'-/5Cy5/GACTTCCTA/TAO/AACAATCTATACAGGTAAT/3IABkFQ/-3' | Cy5 |
| | T22917G | L452R | TGATAGATTTCAGTTGAAATATCTCTCTCA | AATCTTGATTCTAAGGTTGGTGGTAATTAT | 5'-/56-FAM/ACTTCCTAA/ZEN/ACAATCTATACCGGTAAT/3IABkFQ/-3' | FAM |
| P681 | C23604C | P681P | CCCATTGGTGCAGGTATATG | TAGTGTAGGCAATGATGGATTGA | 5'-/5Cy5/ACTCAGACT/TAO/AATTCTCCTCG/3IABkFQ/-3' | Cy5 |
| | C23604A | P681H | CCCATTGGTGCAGGTATATG | TAGTGTAGGCAATGATGGATTGA | 5'-/56-FAM/ACTCAGACT/ZEN/AATTCTCATCG/3IABkFQ/-3' | FAM |
| | C23604G | P681R | CCCATTGGTGCAGGTATATG | TAGTGTAGGCAATGATGGATTGA | 5'-/5HEX/ACTCAGACT/ZEN/AATTCTCGTCG/3IABkFQ/-3' | HEX |

**TABLE 4** Purified gene fragment control sequences

| Nucleotide | Amino acid | Selection sequence | Product length |
|---|---|---|---|
| A23063A | N501N | 5′-TTG TAA TGG TGT TGA AGG TTT TAA TTG TTA CTT TCC TTT ACA ATC ATA TGG TTT CCA ACC CAC TAA TGG TGT TGG TTA CCA ACC ATA CAG AGT AGT AGT ACT TTC TTT TGA ACT TCT ACA TGC ACC AGC AA-3′ | 131 bp |
| A23063T | N501Y | 5′-TTG TAA TGG TGT TGA AGG TTT TAA TTG TTA CTT TCC TTT ACA ATC ATA TGG TTT CCA ACC CAC TTA TGG TGT TGG TTA CCA ACC ATA CAG AGT AGT AGT ACT TTC TTT TGA ACT TCT ACA TGC ACC AGC AA-3′ | 131 bp |
| 21765–21770 | 69-70 wild type | 5′-TCA GTT TTA CAT TCA ACT CAG GAC TTG TTC TTA CCT TTC TTT TCC AAT GTT ACT TGG TTC CAT GCT ATA CAT GTC TCT GGG ACC AAT GGT ACT AAG AGG TTT GAT AAC CCT GTC CTA CCA TTT AAT GAT GG-3′ (WT) | 131 bp |
| G23012G | E484E | 5′-AAC CTT TTG AGA GAG ATA TTT CAA CTG AAA TCT ATC AGG CCG GTA GCA CAC CTT GTA ATG GTG TTG AAG GTT TTA ATT GTT ACT TTC CTT TAC AAT CAT ATG GTT TCC AAC CCA CTA ATG GTG TTG GTT AC-3′ | 132 bp |
| G23012C | E484Q | 5′-TAA TTG TTA CTT TCC TTT ACA ATC ATA TGG TTT CCA ACC CAC TA-3′ | 132 bp |
| G23012A | E484K | 5′-AAC CTT TTG AGA GAG ATA TTT CAA CTG AAA TCT ATC AGG CCG GTA GCA CAC CTT GTA ATG GTG TTA AAG GTT TTA ATT GTT ACT TTC CTT TAC AAT CAT ATG GTT TCC AAC CCA CTA ATG GTG TTG GTT AC-3′ | 132 bp |
| T22917T | L452L | 5′-TGG AAT TCT AAC AAT CTT GAT TCT AAG GTT GGT GGT AAT TAT AAT TAC CTG TAT AGA TTG TTT AGG AAG TCT AAT CTC AAA CCT TTT GAG AGA GAT ATT TCA ACT GAA ATC TAT CAG GCC GGT AGC ACA C-3′ | 130 bp |
| T22917G | L452R | 5′-TGG AAT TCT AAC AAT CTT GAT TCT AAG GTT GGT GGT AAT TAT AAT TAC CGG TAT AGA TTG TTT AGG AAG TCT AAT CTC AAA CCT TTT GAG AGA GAT ATT TCA ACT GAA ATC TAT CAG GCC GGT AGC ACA C-3′ | 130 bp |
| A22812A | K417K | 5′-CTC TGC TTT ACT AAT TC TAT GCA GAT TCA TTT GTA ATT AGA GGT GAT GAA GTC AGA CAA ATC GCT CCA GGG CAA ACT GGA AAG ATT GCT GAT TAT AAT TAT AAA TTA CCA GAT GAT TTT ACA GGC TGC GTT AT-3′ | 134 bp |
| A22812C | K417T | 5′-CTC TGC TTT ACT AAT GTC TAT GCA GAT TCA TTT GTA ATT AGA GGT GAT GAA GTC AGA CAA ATC GCT CCA GGG CAA ACT GGA ACG ATT GCT GAT TAT AAT TAT AAA TTA CCA GAT GAT TTT ACA GGC TGC GTT AT-3′ | 134 bp |
| G22813T | K417N | 5′-CTC TGC TTT ACT AAT GTC TAT GCA GAT TCA TTT GTA ATT AGA GGT GAT GAA GTC AGA CAA ATC GCT CCA GGG CAA ACT GGA AAT ATT GCT GAT TAT AAT TAT AAA TTA CCA GAT GAT TTT ACA GGC TGC GTT AT-3′ | 134 bp |
| C23604C | P681P | 5′-TAT GAG TGT GAC ATA CCC ATT GGT GCA GGT ATA TGC GCT AGT TAT CAG ACT CAG ACT AAT TCT CCT CGG CGG GCA CGT AGT GTA GCT AGT CAA TCC ATC ATT GCC TAC ACT ATG TCA CTT GGT GCA GAA AAT-3′ | 132 bp |
| C23604A | P681H | 5′-TAT GAG TGT GAC ATA CCC ATT GGT GCA GGT ATA TGC GCT AGT TAT CAG ACT CAG ACT AAT TCT CAT CGG CGG GCA CGT AGT GTA GCT AGT CAA TCC ATC ATT GCC TAC ACT ATG TCA CTT GGT GCA GAA AAT-3′ | 132 bp |
| C23604G | P681R | 5′-TAT GAG TGT GAC ATA CCC ATT GGT GCA GGT ATA TGC GCT AGT TAT CAG ACT CAG ACT AAT TCT CGT CGG CGG GCA CGT AGT GTA GCT AGT CAA TCC ATC ATT GCC TAC ACT ATG TCA CTT GGT GCA GAA AAT-3′ | 132 bp |

**VOC assay accuracy and analytical specificity.** To verify the assay accuracy in our RT-PCR method, VOCs were also detected used VirSNiP SARS-CoV-2 mutation assays for strain surveillance (TIB Molbiol, Berlin, Germany), which used real-time RT-PCR postmelting curve analysis to detect mutations targeting specific spike protein variations (HV69/70, K417N, L452R, E484K, N501Y, P681H, and P681R). These assays were performed as described previously (27). To testify the analytical specificity of the VOC assay, we obtained samples of known upper respiratory viruses, including influenza A, influenza B, rhinovirus, enterovirus, parainfluenza virus subtypes 1 to 3, and adenovirus as viral cultures from the Taiwan CDC viral infection contract laboratory. Additionally, 500 COVID-19-negative specimens were used to evaluate the analytical specificity of the lab-developed multiplex PCR VOC test performance.

**Analytical validation using gene fragments control.** We used purified gene fragments controls (gBlocks, Integrated DNA Technologies, IDT) of SARS-CoV-2 viral genes for absolute quantification (Table 4). These controls were used to prepare a serial dilution panel with approximately 20 replicates. We prepared dilutions of the controls (7.5, 15, 30, 60, 120, and 240 copies/$\mu$L) using nuclease-free water to assess the limit of detection (LoD) on two different thermocyclers (LightCycler 96 and LabTurbo AIO 48 system). LoD was defined as 95% probability of positive replicates.

**Data availability.** Data are available upon request.

## SUPPLEMENTAL MATERIAL

Supplemental material is available online only.

**SUPPLEMENTAL FILE 1**, PDF file, 0.2 MB.

## ACKNOWLEDGMENTS

Conceptualization: Chih-Hung Wang, Jung-Chung Lin, Kuo-Ming Ye, Chien-Wen Chen, and Feng-Yee Chang; data curation: Ming-Jr Jian and Hsing-Yi Chung; formal analysis: Ming-Jr Jian and Kuo-Sheng Hung; investigation: Hsing-Yi Chung, Chih-Kai Chang, and Cherng-Lih Perng; methodology: Chih-Kai Chang, Shan-Shan Hsieh, Kuo-Sheng Hung, and Cherng-Lih Perng; supervision: Chih-Hung Wang, Hung-Sheng Shang, Jung-Chung Lin, Kuo-Ming Ye, Chien-Wen Chen, Sheng-Hui Tang, and Feng-Yee Chang; writing – original draft: Ming-Jr Jian; writing – review & editing: Ming-Jr Jian and Hung-Sheng Shang. All authors made substantial contributions to conception and design, acquisition of data, and/or analysis and interpretation of data, took part in drafting the article or revising it critically for important intellectual content, agreed to submit to the current journal, gave final approval of the version to be published, and agree to be accountable for all aspects of the work.

This study was supported by the Tri-Service General Hospital, Taipei, Taiwan, ROC (grant number TSGH-D- 111086). The funders had no role in study design, data collection and interpretation, or the decision to submit the work for publication.

This study was approved by the Institutional Review Board of Tri-Service General Hospital (TSGHIRB No.: C202005041), registered on Feb 8, 2021.

Informed consent was obtained from all subjects involved in the study.

We declare no conflict of interest.

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
