## [Reviewer comments · Microbiology Spectrum]

Microbiology Spectrum

Emergency SARS-CoV-2 Variants of Concern: Novel Multiplex real-time RT-PCR Assay for Rapid Detection and Surveillance

Hsing-Yi Chung, Ming-Jr JIAN, Chih-Kai Chang, Jung-Chung Lin, Kuo-Ming Yeh, Chien-Wen Chen, Shan-Shan Hsieh, Kuo-Sheng Hung, Sheng-Hui Tang, Cherng-Lih Perng, Feng-Yee Chang, Chih-Hung Wang, and Hung-Sheng Shang

Corresponding Author(s): Hung-Sheng Shang, Tri-Service General Hospital

Review Timeline:

Submission Date:	December 4, 2021
Editorial Decision:	January 5, 2022
Revision Received:	January 28, 2022
Accepted:	February 3, 2022

Editor: Tulip Jhaveri

Reviewer(s): Disclosure of reviewer identity is with reference to reviewer comments included in decision letter(s). The following individuals involved in review of your submission have agreed to reveal their identity: Yao Jiang (Reviewer #1); Luciana Jesus Costa (Reviewer #2)

Transaction Report:

DOI: <https://doi.org/10.1128/spectrum.02513-21>

January 5, 2022

Dr. Hung-Sheng Shang
Tri-Service General Hospital
Division of Clinical Pathology, Department of Pathology
No. 325, Sec.2, Chenggong Road, Neihu District, Taipei City, Taiwan
Taipei
Taiwan

Re: Spectrum02513-21 (Emergency SARS-CoV-2 Variants of Concern: Novel multiplex real-time RT-PCR Assay for Rapid Detection and Surveillance)

Dear Dr. Hung-Sheng Shang:

Link Not Available

Sincerely,

Tulip Jhaveri

Journals Department
Reviewer comments:

Reviewer #1 (Comments for the Author):

The study established a multiplex PCR typing method to detect 9 mutations with specific primers and probes (Δ HV 69-70, K417T, K417N, L452R, E484K, E484Q, N501Y, P681H and P681R) against the receptor-binding domain of the spike protein of SARS-CoV-2 variants. This PCR typing strategy allowed the detection of four major variants of concern, and also provided an open-source PCR assay which could rapidly be deployed in laboratories, to enhance surveillance for the local emergence and spread of B.1.1.7, B.1.351, P.1, and B.1.617.2 variants as well as four Omicron mutations on the Spike protein (Δ HV 69/70, N501Y, K417N, P681H).

1. Authors only collected positive nasopharyngeal swab specimens, lack of negative samples. It deserves to explore the specificity of this PCR typing method.

2. Please provide corresponding references about setting the threshold (Ct) value.

3. In the Figure 3, authors divided samples into three categories by VOC testing, namely N501Y (+), P681R (+), and N501Y (-), P681R (-) groups. They thought that N501Y (+) group concluded the Alpha, Beta, Gamma variants. In fact, it is unreasonable to

classify these variants only depending on one mutation (N501Y).

4. Is there a difference between this multiplex PCR typing method and other current detecting methods for SARS-CoV-2 and variants? It is necessary to compare their differences using a table.

Reviewer #2 (Comments for the Author):

The manuscript by Chung and co-workers describes the feasibility of using RT-qPCR as a method to detect SARS-CoV-2 VOCs currently circulating. This is a very important issue due to the continuous emergence of variants that impact the course of pandemic as we witnessed not long before with the Delta and now with the Omicron variant. New sensitivity and reliable alternatives to determine variants other than NGS are urgently required to speed up variant determination which in the case of the Omicron variant will be important in terms of monoclonal treatment for instance. In this scenario, this paper is relevant, and data described here points out to the possibility of using multiplex RT-qPCR for such a determination. However, the manuscript fails in describing the methodology in more details. When authors call the assay multiplex is confusing since from the way they describe the method it is not possible to VOC? Thus, a comprehensive description of the methodology should be provided. Also, authors should provide bulk data for the tests performed, mostly in terms of Ct range for each set of primers and probe and how they relate to each other. Importantly, authors should provide a summary from sequencing results (or representative sequences). It would be nice to have more information regarding how Omicron will perform in this assay.

On a minor note, text needs a grammatical revision since in several parts sentences are lacking the correct grammatical construction.

In figure 1 is hard to see the difference between the arrow bar and the long bar.

Staff Comments:

Preparing Revision Guidelines

Please return the manuscript within 60 days; if you cannot complete the modification within this time period, please contact me. If you do not wish to modify the manuscript and prefer to submit it to another journal, please notify me of your decision immediately so that the manuscript may be formally withdrawn from consideration by Microbiology Spectrum.

The study established a multiplex PCR typing method to detect 9 mutations with specific primers and probes (Δ HV 69-70, K417T, K417N, L452R, E484K, E484Q, N501Y, P681H and P681R) against the receptor-binding domain of the spike protein of SARS-CoV-2 variants. This PCR typing strategy allowed the detection of four major variants of concern, and also provided an open-source PCR assay which could rapidly be deployed in laboratories, to enhance surveillance for the local emergence and spread of B.1.1.7, B.1.351, P.1, and B.1.617.2 variants as well as four Omicron mutations on the Spike protein (Δ HV 69/70, N501Y, K417N, P681H).

1. Authors only collected positive nasopharyngeal swab specimens, lack of negative samples. It deserves to explore the specificity of this PCR typing method.
2. Please provide corresponding references about setting the threshold (Ct) value.
3. In the Figure 3, authors divided samples into three categories by VOC testing, namely N501Y (+), P681R (+), and N501Y (-), P681R (-) groups. They thought that N501Y (+) group concluded the Alpha, Beta, Gamma variants. In fact, it is unreasonable to classify these variants only depending on one mutation (N501Y).
4. Is there a difference between this multiplex PCR typing method and other current detecting methods for SARS-CoV-2 and variants? It is necessary to compare their differences using a table.

[2022.01.26]

Dr. Christina Cuomo
Editor-in-Chief

Re: Resubmission of Spectrum02513-21

Dear Editor:

Thank you very much for your decision letter and review of our manuscript titled "**Emergency SARS-CoV-2 Variants of Concern: Novel multiplex real-time RT-PCR Assay for Rapid Detection and Surveillance**". We are grateful for the helpful and clear comments received and for the opportunity to submit a revised version of the manuscript. We have taken all of the reviewer comments and the editor evaluation into account in this revision. Changes are highlighted in yellow in the manuscript, and a "clean" copy without highlighting is also included for further review. Please also find attached a point-by-point response to each of the reviewer comments.

We hope that our responses and revisions are satisfactory, and that the manuscript is now deemed suitable for publication in *Microbiology Spectrum*.

Thank you for your continued consideration. We look forward to hearing from you.

Sincerely,
Hung-Sheng Shang
Division of Clinical Pathology,
Department of Pathology,
Tri-Service General Hospital,
National Defense Medical Center
Taipei, Taiwan
iamkeith001@gmail.com

Chih-Hung Wang
Department of Otolaryngology-Head and Neck Surgery
Tri-Service General Hospital
National Defense Medical Center
Taipei, Taiwan
E-mail: chw@ms3.hinet.net

Reviewer #1 (Comments for the Author):

The study established a multiplex PCR typing method to detect 9 mutations with specific primers and probes (Δ HV 69-70, K417T, K417N, L452R, E484K, E484Q, N501Y, P681H and P681R) against the receptor-binding domain of the spike protein of SARS-CoV-2 variants. This PCR typing strategy allowed the detection of four major variants of concern, and also provided an open-source PCR assay which could rapidly be deployed in laboratories, to enhance surveillance for the local emergence and spread of B.1.1.7, B.1.351, P.1, and B.1.617.2 variants as well as four Omicron mutations on the Spike protein (Δ HV 69/70, N501Y, K417N, P681H).

1. Authors only collected positive nasopharyngeal swab specimens, lack of negative samples. It deserves to explore the specificity of this PCR typing method.

Response: We agree with your suggestion. We added 500 COVID-19-negative specimens and samples of known upper respiratory viruses to this study. We have edited the manuscript in lines 124-132 and 258-263 to describe the specificity of our PCR typing method

2. Please provide corresponding references about setting the threshold (Ct) value.

Response: We thank the reviewer for this comment. We have edited lines 118-120 and added Supplementary Table 1 to provide the setting the threshold (Ct) value for the VOC typing assay. We also added line 223-226 to define the positive results in SARS-CoV-2 about setting the Ct value.

3. In the Figure 3, authors divided samples into three categories by VOC testing, namely N501Y (+), P681R (+), and N501Y (-), P681R (-) groups. They thought that N501Y (+) group concluded the Alpha, Beta, Gamma variants. In fact, it is unreasonable to classify these variants only depending on one mutation (N501Y).

Response: We thank the reviewer for this comment, and have edited Figure 3 accordingly.

4. Is there a difference between this multiplex PCR typing method and other current detecting methods for SARS-CoV-2 and variants? It is necessary to compare their differences using a table.

Response: We thank the reviewer for pointing this out. We have edited lines 141-145 and added Table 3 to compare the accuracy of our PCR typing method and that of commercial kits.

Reviewer #2 (Comments for the Author):

The manuscript by Chung and co-workers describes the feasibility of using RT-qPCR as a method to detect SARS-CoV-2 VOCs currently circulating. This is a very important issue due to the continuous emergence of variants that impact the course of pandemic as we witnessed not long before with the Delta and now with the Omicron variant. New sensitivity and reliable alternatives to determine variants other than NGS are urgently required to speed up variant determination which in the case of the Omicron variant will be important in terms of monoclonal treatment for instance. In this scenario, this paper is relevant, and data described here points out to the possibility of using multiplex RT-qPCR for such a determination.

1. However, the manuscript fails in describing the methodology in more details. When authors call the assay multiplex is confusing since from the way they describe the method it is not possible to VOC? Thus, a comprehensive description of the methodology should be provided.

Response: We thank the reviewer for this suggestion. We have included a comprehensive description of the methodology in the Results (lines 115-132 and 141-145) and Methods (lines 212-214, 223-225, and 252-263) sections.

2. Also, authors should provide bulk data for the tests performed, mostly in terms of Ct range for each set of primers and probe and how they relate to each other.

Response: We agree with this suggestion. We have edited Table 1 and lines 115-123, and have added Supplementary Table 1 to provide bulk data for the tests performed in terms of Ct range.

3. Importantly, authors should provide a summary from sequencing results (or representative sequences). It would be nice to have more information regarding how Omicron will perform in this assay.

Response: We thank the reviewer for this comment. We have added lines 141-145 and Table 3 to compare the test accuracy of our PCR typing method with that of commercial kits. Also, we have added Supplementary Table 2 to address Omicron PCR typing performance compared with that of Whole genome sequencing (line 194-197). The VOC typing methods in our study showed consistent results.

4. On a minor note, text needs a grammatical revision since in several parts sentences are lacking the correct grammatical construction.

Response: We thank the reviewer for pointing this out. We have carefully corrected the typo and revised the entire manuscript for grammatical errors.

5. In figure 1 it is hard to see the difference between the arrow bar and the long bar.

Response: We agree, but believe that the reviewer may be referring to Figure 2. We have edited Figure 2 such that readers can clearly see the difference between the arrow bar and the long bar.

February 3, 2022

Dr. Hung-Sheng Shang
Tri-Service General Hospital
Division of Clinical Pathology, Department of Pathology
No. 325, Sec.2, Chenggong Road, Neihu District, Taipei City, Taiwan
Taipei
Taiwan

Re: Spectrum02513-21R1 (Emergency SARS-CoV-2 Variants of Concern: Novel Multiplex real-time RT-PCR Assay for Rapid Detection and Surveillance)

Dear Dr. Hung-Sheng Shang:

Your manuscript has been accepted, and I am forwarding it to the ASM Journals Department for publication. You will be notified when your proofs are ready to be viewed.

Sincerely,

Tulip Jhaveri
Editor, Microbiology Spectrum
